A new model for simulating growth in fish

Hamre Johannes 1
Johnsen Espen 1
Hamre Kristin 2 kha@nifes.no
1 Institute of Marine Research , Bergen , Norway
2 National Institute of Nutrition and Seafood Research (NIFES) , Bergen , Norway
Song Linsheng
Electronic publication date: 2014 Jan 30
Publication date: 2014
Volume: 2
Electronic Location ID: e244
Received 2013 Sep 23; Accepted 2013 Dec 21
Copyright: © 2014 Hamre et al.
Copyright year: 2014
Copyright holder: Hamre et al.
License: This is an open access article distributed under the terms of the Creative Commons Attribution License, which permits unrestricted use, distribution, and reproduction in any medium, provided the original author and source are credited.
License URL: https://creativecommons.org/licenses/by/3.0/

Keywords: Growth model, Fish growth, Vertebrate growth, Fish stock assessment, Fisheries management

Funding: The hours used for this study were covered by internal funds in our respective institutions. The funders had no role in study design, data collection and analysis, decision to publish, or preparation of the manuscript.

==============================
A real dynamic population model calculates change in population sizes independent of time. The Beverton & Holt (B&H) model commonly used in fish assessment includes the von Bertalanffy growth function which has age or accumulated time as an independent variable. As a result the B&H model has to assume constant fish growth. However, growth in fish is highly variable depending on food availability and environmental conditions. We propose a new growth model where the length increment of fish living under constant conditions and unlimited food supply, decreases linearly with increasing fish length until it reaches zero at a maximal fish length. The model is independent of time and includes a term which accounts for the environmental variation. In the present study, the model was validated in zebrafish held at constant conditions. There was a good fit of the model to data on observed growth in Norwegian spring spawning herring, capelin from the Barents Sea, North Sea herring and in farmed coastal cod. Growth data from Walleye Pollock from the Eastern Bering Sea and blue whiting from the Norwegian Sea also fitted reasonably well to the model, whereas data from cod from the North Sea showed a good fit to the model only above a length of 70 cm. Cod from the Barents Sea did not grow according to the model. The last results can be explained by environmental factors and variable food availability in the time under study. The model implicates that the efficiency of energy conversion from food decreases as the individual animal approaches its maximal length and is postulated to represent a natural law of fish growth.

Introduction

A main objective in fisheries management is to track the impact of fisheries on the fish stocks and predict the maximum sustainable yield (MSY) (Pitcher & Hart, 1982). A real dynamic population model calculates changes in population size and biomass independent of time. In mathematical terms, such a model is often formulated with differential equations where the change is initially measured as a function of time. The time factor is then removed by integration, but the differential models describing increment in biomass of fish cannot be integrated (Beverton & Holt, 1957; Lotka, 1925; Schaefer, 1957; Volterra, 1926). Furthermore, fish growth, i.e., the size increment with time, varies greatly with food quality and availability, temperature and other environmental factors and the fish will reach the different stages in development more dependent on size than on age (Amara & Lagardere, 1995; Aritaki & Seikai, 2004; Sæle & Pittman, 2010). A certain size or energy store is needed for metamorphosis in fish larvae (Amara & Lagardere, 1995; Aritaki & Seikai, 2004), smoltification in salmon (McCormick & Bjørnsson, 1994) and sexual maturation in fish in general. Accordingly, farmed cod grow faster and mature at an earlier age than wild cod (Braaten, 1984; Karlsen, Holm & Kjesbu, 1995; Karlsen et al., 2006). If the required size is not obtained, the fish will simply postpone development. Therefore, a growth function which omits time and is based on fish size would be in line with real fish growth and development, and would be preferred for calculation of yield in simulation models.

Population dynamic models built on the classical Beverton & Holt (B&H) model are often used in fish stock assessments (Beverton & Holt, 1957; Hilborn, 1994). This model estimates the yield per recruit by assuming that growth is a function of age, e.g., accumulated time in line with von Bertalanffy (1938). As a result, the B&H model cannot sum up the yield in all year classes in the same year, but it can be used to sum up the yield in one year-class during life. The authors therefore assume that the yield of all year-classes in one year is equivalent to the yield of one year-class through life (Beverton & Holt, 1957), e.g., the model assumes constant growth.

Another way to quantify changes in mass than using differential equations was introduced by Albert Einstein, when he presented the theory of relativity. Instead of using mathematics to deduce from a known physical law, he postulated that energy and mass are equivalents and depend on the speed of light (Einstein, 1905). Einstein’s postulate was effectively proven by the testing of the nuclear bombs and by later experiments with particle acceleration. Our postulate is that the length increment (dL) is inversely proportional to the length (Ls) and that dL is reduced towards a maximum length (Lmax): (1) dL=k⋅(Lmax−Ls)

k is a variable determined by environmental factors, such as food availability and temperature. Equation (1) includes only measurable factors so that dL can be summed arithmetically, either per year or by year classes, using computer techniques. It can be applied for vertebrates, only, because the dynamic term (Lmax − Ls) cannot be negative, i.e., the animal cannot shrink.

The present study aims at developing a real dynamic growth model of fish biomass based on an unchangeable and predictable natural law which is independent of time. Since environmental conditions modulate growth, we used an experiment with zebrafish (Danio reirio) kept at constant conditions and fed Artemia for one generation (Gomez-Requeni et al., 2010) to validate the growth model. Another experiment, where zebrafish were fed a formulated diet from first-feeding until sexual maturation, also under near constant environmental conditions (Kaushik, Georga & Koumoundouros, 2011), is included. We also investigate how the model fits to the observed growth of Norwegian spring spawning herring (Clupea harengus), herring and blue whiting (Micromesistius poutassou) in the Norwegian Sea, North East Atlantic capelin (Mallotus villosus) and cod (Gadus morhua), cod from the North Sea, farmed cod originating from a Norwegian costal stock and Walleye Pollock (Theragra chalcogramma) from the North Pacific. Later we intend to use this law to simulate the development in fish stocks, starting with measurement of the state of the stocks using acoustic surveys, in accordance with Bjerknes’ principle of exact science (Bjerknes, 1904).

Materials and methods

Data acquisition and organisation

The data on growth in zebrafish were extracted from Gomez-Requeni et al. (2010) and Kaushik, Georga & Koumoundouros (2011). Briefly, in the first experiment, the zebrafish were held in 50 5 L tanks at 8 fish per tank from 16 until 103 dpf (days post fertilization) and fed Artemia nauplii as the only diet from 16 dpf. Before 16 dpf, a dry commercial diet was fed. Photoperiod was 13L:11D (light:dark), and water temperature was constantly kept at 28°C. Mean values for pH and conductivity were 7.21 and 685 μS, respectively. At regular intervals, approximately 30 fish were sacrificed and fork length and body weight were measured. In the second zebrafish experiment, the fish were held in 10 L aquaria which were connected to a water recirculation system. The fish density was initially 350 fish per tank and reduced to 150 fish per tank when the fish reached 15 mm total length (TL). Two days after hatching, triplicate groups of fish started receiving a dry micro-particulate feed of commercial origin (Gemma micro; Skretting). Feed was distributed by hand to fish four times daily to visual satiety until circa 12 mm TL and two to three times daily thereafter. The pellet size was increased as the fish grew. Temperature was maintained at 28°C, pH at 7.0–7.5, oxygen saturation at 70%–95%, and photoperiod at 14L:10D (light:dark). 8–10 fish were taken for measurement of TL at regular intervals.

Mean lengths (L) of Norwegian spring spawning herring, and North Atlantic capelin and cod were from ICES working group reports (ICES, 2012a, ICES, 2012c). In herring, data on 1 to 9 year old fish, from 1985 until 2003 were used, while fish older than 9 years were omitted from the dataset, because they had essentially no length increment. The data were organized as development of length with age in different year classes. Full year classes were obtained from those reaching one year of age in 1985-1995. The North Sea herring data are given as the mean length at age measured during the acoustic surveys in June–July 2002–2012, and collected from the ICES herring assessment working groups for the area south of 62°N (ICES, 2013). Data on capelin and cod from the Barents Sea were also organized as development of length per year-class, the 1986–2001 year-classes for capelin and the 1985–1995 year-classes for cod. Data on weights of North Sea cod were taken from the ICES working group report (ICES, 2012b) and converted to length using a condition factor of 0.0104 (Daan, 1974). The data on farmed cod were from Norwegian coastal cod hatched at Parisvannet and reared in 5×5×5 m3 pens at Austevoll Aquaculture Research Station, Institute of Marine Research, Bergen Area, Norway. All fish were fed in moderate excess with commercial dry pellets (Tess Marine, Skretting AS, Stavanger, Norway; 49% protein, 7.1% lipid and 33.5% carbohydrate). The temperature measured at 2 m depth varied between approximately 14°C in July–September to 5°C in February–April and the salinity varied between 28 and 32 g/L. The fish became sexually mature at 2 years of age (Lehmann et al., 1990; Kjesbu & Holm, 1994). Pollock data, which are collected during bottom trawl surveys, were kindly communicated by Stan Kotwicki at the Alaska Fisheries Science Center, National Marine Fisheries Service. The Pollock stock is described in Ianelli et al. (2011). Blue whiting data are collected from Norwegian landings west of the British Isles where the main fishery takes place in February–April. The data were kindly provided by Kjell Utne Rong, Institute of Marine Research, Norway. Both Pollock and blue whiting data were organized as development of length per year class, for the 1982–2000 and 1982–2004 year classes, respectively.

Calculations and statistics

Increment in length (dL) was calculated as dL = L(a+1) − L(a), where a is fish age, using time intervals of one day for zebrafish, one year for farmed cod and for the wild fish stocks.

The statistical treatment was performed with GraphPad Prism, ver. 6 for Windows (GraphPad Inc., La Jolla, CA, USA). For the zebrafish data of Gomez-Requeni et al. (2010), the plot of length increment day−1 by length was first fitted to a second order polynomial equation. The points after the maximum of the obtained parable were then fitted to a linear and a second order polynomial equation and the results compared using the GraphPad software. The zebrafish data of Kaushik, Georga & Koumoundouros (2011) of length increment by length were fitted to a linear equation. For herring, capelin, wild and farmed cod, average length increment year−1 by length in the year-classes were also fitted to a linear and a second order polynomial equation, and the results compared by the software.

For Norwegian spring spawning herring, length increment year−1 for 1–9 year old fish could be fitted to a linear equation in some of the year classes, while in other year classes, one and two year old fish had lowered length increment. In all cases, R2 > 0.98 for 3–9 year old fish and Lmax could be calculated from these data (average Lmax = 354 mm), while the slope of the line equals k in this period. k in one and two year old fish, when they deviated from the linear relationship, was calculated as the slope of the straight line between length increment year−1 in these fish and Lmax.

Results

In the experiment of Gomez-Requeni et al. (2010), zebrafish growth as a function of age fits a sigmoid curve (Fig. 1A), and plotting length increment as a function of length gave an almost perfect fit to the second order polynomial y = −0.0027x2 + 0.078x + 0.25 (R2 = 0.994, Fig. 1B). When omitting the first two points corresponding to the larval and very early juvenile stages, and comparing the polynomial (R2 = 0.9988) and the linear equation, the second order polynomial was preferred (p = 0.015), but the straight line still had an R2 = 0.9887 (Fig. 1C). Therefore, growth in zebrafish held at constant conditions after the juvenile stage can be formulated mathematically by (1): dL=k⋅(Lmax−Ls)=−k⋅Ls+k⋅Lmax.

The slope of the curve corresponds to − k, while Lmax can be calculated from the intercept with the y-axis; k ⋅ Lmax (Fig. 1C).

Figure 1 Growth in zebrafish kept under constant conditions and fed to satiation through one generation.

(A) Growth in length by age (mean ± SD, Gomez-Requeni et al. (2010), n = 30, Kaushik, Georga & Koumoundouros (2011), n = 10–18). (B) Calculated length increment by length for all data-points in Gomez-Requeni et al. (2010) with a second order polynomial fit (y = −0.0027x2 + 0.078x + 0.25 (R2 = 0.994). (C) Points past the exponential growth phase in Gomez-Requeni et al. (2010) fitted to a 2nd order polynomial (blue line, R2 = 0.999) and a linear (red line, R2 = 0.989) equation and data from Kaushik, Georga & Koumoundouros (2011), fitted to a linear (R2 = 0.55) or an exponential (R2 = 0.60) equation (green; the lines of the models are overlapping).

The experiment of Kaushik, Georga & Koumoundouros (2011), with zebrafish, showed a less regular growth curve as length by age (Fig. 1A). Fitting (1) to the data still gave an R2 = 0.55, the fit to the 2nd order polynomial equation han an R2 of 0.60, but was rejected (p = 0.58). The slopes of the lines were not different, but the difference in Lmax between the two zebrafish groups was highly significant (p = 0.0007, Fig. 1C).

Figure 2 Growth in the year-classes 1985–1995 of Norwegian spring spawning herring.

(A) Growth as length by age (mean ± SD, n =11 year classes). (B) Average length increment by length in all year-classes from 1985 until 1995, fitted to a 2nd order polynomial (blue line, R2 = 0.998) and a linear (red line, R2 = 0.990) equation. (C) Length increment by length fitted to linear equations in the year-classes 1988 (R2 = 0.996) and 1989 (R2 = 0.996). (D) Length increment by length in the year-class 1993, where the points for 3–9 year old fish were fitted to a linear equation (R2 = 0.996).

The curve of length by age of the 1985 to the 1995 year-classes of herring from 1 until 9 years of age (Fig. 2A) shows rapid growth in length of young fish and a decrease in length increment as the fish get older and increase in size. The decrease in length increment starts already after the second year and lasts until year 9, where-after the length increment per year is close to zero (data not shown). Figure 2B shows the plot of average length increment by length in all year-classes. The fit to a second order polynomial has an R2 = 0.9979, while the linear fit has an R2 = 0.9904. Here the second order equation is again preferred (p = 0.01). Figure 2C and 2D shows that the plots of length increment by length differ in the different year classes. In 1988 and -89, length increment through the whole life cycle fitted a linear equation (Fig. 2C), while in 1993, length increment in one and two year old fish was low compared to that in 1988 and 89 and deviated from the linear equation obtained for 3–9 year old fish. The length increment for 1 and 2 year old fish deviated from the straight line obtained for in 3–9 year old fish (R2 > 0.98) in the year-classes 1985, -86, -92, -93 and -94. The length increment in 1 year old fish deviated from the straight line obtained for 2–9 year old fish (R2 = 0.99) in the year-class 1987. All ages fitted a straight line in the 1988, -89, -90, -91 and -95 year-classes (R2 > 0.98). Assuming that the deviations from the linear relationship are due to differences in environmental conditions that can be expressed by k, k was plotted by year-class for 1 year old, 2 year old and 3–9 year old herring, respectively, from 1985 to 1995 (Fig. 3).

Figure 3 k in herring of different ages in the year-classes 1985–1995.

k was calculated from the slope of (1): dL = k ⋅ (Lmax − Ls), where dL is length increment and Ls is the measured length.

Growth in capelin given as length by age is shown in Fig. 4A. As in herring, the length increment decreases as the fish increase in size, however the number of points in the regression is limited by the life span of capelin, which is only 4–5 years, since almost all fish die after the first spawning. Average length increment by length in the year-classes from 1986 to 2001 is given in Fig. 4B with a 2nd order polynomial fit (R2 = 0.9983) and a linear fit (R2 = 0.9766). In this case, the linear equation was preferred (p = 0.17).

Figure 4 Growth in capelin from the Barents Sea.

(A) Growth as length by age (Mean ± SD, n = 16 year classes). (B) Average length increment by length in all year-classes from 1986 until 2001, fitted to a 2nd order polynomial (blue line, R2 = 0.998) and a linear (red line, R2 = 0.977) equation.

Figure 5 shows the age and length dependent growth of North Sea herring, blue whiting from the Norwegian Sea and Walleye Pollock from the Eastern Bering Sea. In North Sea herring (Fig. 5A and 5B), (1) and the 2nd order polynomial model did not give different R2 (0.97 and 0.98, respectively, p = 0.36). In blue whiting (Fig. 5B and Fig. 5C), the R2 were 0.89 and 0.98 for the linear versus the polynomial model and the polynomial model was preferred (p = 0.02). A similar situation was found in Pollock (Fig. 5D and 5E) where R2 were 0.95 and 0.99, respectively, and again a polynomial model was preferred (p = 0.0002). For both blue whiting and Pollock, the constant of the first order term of the polynomial model was negative, inverting the curve compared to the polynomial models fitted to growth data of the other species.

Figure 5 Growth in North Sea herring, blue whiting from the Norwegian Sea and Walleye Pollock from the Eastern Bering Sea.

(A), (C), (E) Growth as length by age (mean ± SD, n = 11 survey years for North Sea herring, n = 22 year classes for blue whiting and n = 18 year classes for Pollock). (B), (D), (F) Average length increment by length in all year-classes listed above, fitted to a 2nd order polynomial (blue line, R2 = 0.977, 0.980, 0.988, for herring, blue whiting and Pollock, respectively) and a linear (red line, R2 = 0.973, 0.888, 0.948) equation.

Growth in Northeast Arctic cod of different year-classes, given as length by age, seems to fit a linear equation, not a sigmoid curve as in the other fish stocks (Fig. 6A). Average length increment by length did not decrease with size to the same extent as in the other fish species, varying between 11 and 8 cm year−1 (Fig. 6B). Regression analyses using data from individual year classes gave variable and sometimes poor relationships, with R2 = 0.85–0.27 for the second order and R2 = 0.88–0.06 for the linear relationship. Growth in North Sea cod given as length by age (Fig. 6C), did follow a sigmoid curve. Length increment by length (Fig. 6D) showed a good fit to (1) after the fish had reached a length of 70 cm (R2 = 0.98). The 2nd order polynomial fit for this period was rejected (R2 = 0.98, p = 1.0). Before the fish reached 70 cm, length increment was lowered compared to what is predicted by (1). Growth in length by age in farmed cod kept at relatively constant conditions and fed a dry diet to satiation, followed an asymptotic curve, length increment by length in showed a good fit to (1) (Fig. 6E, R2 = 0.85) and the linear function was preferred over the second order one (p = 0.32).

Figure 6 Growth in cod.

(A) Barents Sea cod. Growth as length by age (Mean ± SD, n = 11 year classes). (B) Barents Sea cod. Average length increment by length in all year-classes from 1985 until 1995. (C) North Sea cod. Growth as length by age (Mean ± SD, n = 11 year classes). (D) North Sea cod. Average length increment by length in all year-classes from 1985 until 1995. The points for fish longer than 70 cm are fitted to an 2nd order polynomial (R2 = 0.980) and a linear (R2 = 0.979) equation (both equations are represented by the red line). (E) Farmed coastal cod held in net pens with seasonal variation in light and temperature but with constant feed supply, feed and feeding regime for one generation. Growth as length by age (Mean ± SD, n = 70 fish). (F) Farmed coastal cod. Average length increment by length fitted to an 2nd order polynomial (blue, R2 = 0.801) and a linear (red, R2 = 0.788) equation.

Discussion

The questions asked in this study are: (1) Do fish grow according to an unchangeable natural law, independent of time? (2) Can such a law be described by (1)? The first question cannot be proved using mathematical deduction, but we postulate a law in a similar way as Albert Einstein postulated the theory of relativity. Question 2 is validated by referring to a model organism for mechanistic biological research, zebrafish.

The zebrafish from which the present growth curves were extracted had been held at constant environmental conditions and fed Artemia, only, from 16 dpf, or a formulated diet from 5 dpf, according to standard feeding regimes, until the fish were past sexual maturation (Gomez-Requeni et al., 2010; Kaushik, Georga & Koumoundouros, 2011). The first experiment created a conventional sigmoid curve of weight by age, which seems to be common in all terrestrial vertebrates (Dumas, France & Bureau, 2010). According to Dumas, France & Bureau (2010) and references therein, there is doubt that fish approach a maximum weight, since growth continues after sexual maturation. In our opinion, the data presented here strongly indicate that an Lmax exists for a number of fish species and that varying environmental conditions can explain why an Lmax appears less evident in some of the species.

In order to omit time from the growth equation, length increment was plotted as a function of length. Regarding the data on zebrafish growth from Gomez-Requeni et al. (2010), this plot had an almost perfect fit to a second order equation, e.g., length increment increased in larval and early juvenile fish (9–30 dpf), approaching a maximum, and decreased gradually in larger fish, reaching zero at a maximal length (Lmax). In the experiment of Kaushik, Georga & Koumoundouros (2011) length increment in 13–20 dpf fish, was higher than length increment in older fish. This may indicate that the lower growth in young fish in the first experiment had to do with a concomitant shift from a dry diet to Artemia at 16 dpf. Artemia are too big to be eaten by first-feeding larvae, and it may be that the fish still needed some time after the diet shift before it could utilize Artemia at maximal efficiency. In the second experiment the fish had been fed the same formulated diet from first-feeding at 5 dpf until the end of the experiment, only pellet size being adjusted to the fish size. The high growth of the early larvae in this experiment, where there was no change of feed, indicates that (1) is also valid also for young zebrafish. The large variation in length increment relative to the regression line may be explained by the lower number of fish measured per point and/or the difficulty in feeding a formulated diets to small fish. Length increment leads to accumulation of biomass and energy. Therefore, the model implies that energy in the food is used most efficiently for biomass accumulation in relatively small fish, with decreasing efficiency as the fish approach Lmax. The linear model given in (1) can also be seen as a simple version of the second half of the 2nd order growth function. It fits well to the data for zebrafish past maximum length increment in the first experiment, even though it is not statistically preferred over the 2nd order equation. Equation (1) is easy to understand, it can readily be adapted in simulation models and it includes the variable k, which describes changes in the environment and food availability. Furthermore, the assessment of fish stocks most often only includes fish that are larger than the point where maximal length increment occurs in the experiment of Gomez-Requeni et al. (2010), again favoring the use of (1).

Equation (1) also fits the length increment by length data for the year-classes of fish, one year old or more, of North Sea herring, capelin and farmed cod. The data from Norwegian Spring Spawning herring of 3–9 years are very well described by (1), but in some of the year-classes, one and two year old fish grow slower than the prediction of (1). The deviation from the straight line is present in some year-classes, but not in others. Therefore we have interpreted it as a difference in environmental conditions, e.g., a difference in k. The cause of a lowered k in young herring may be that they grow up in the colder Barents Sea and move to the Norwegian Sea when they are 20–22 cm, usually about three years old (Holst, Røttingen & Melle, 2004).

Length increment by length in cod from the Barents Sea shows a variable and generally poor fit to (1). This may be caused by low and varying food availability, since the Northeast Arctic cod stock is large and may therefore encounter food shortage (Jørgensen, 1992). The stock also encounters temperatures that are near its minimum tolerance level, which would have a large effect on growth (Dumas, France & Bureau, 2010). Furthermore, older and larger Northeast Arctic cod have a tendency to distribute further to the west, where they encounter gradually increasing temperatures (Michalsen, Ottersen & Nakken, 1998). The slow reduction of length increment when length increases in Northeast Arctic cod, may therefore be explained by a temperature dependent increase in k from one year to the next over the whole life span. Unfortunately, the data from Northeast Arctic cod cannot be used to determine Lmax, so that k cannot be defined per year.

The farmed cod were held under semi-controlled conditions, in net pens with ambient sea-water temperature and natural light, but were fed the same diet using the same feeding regime for one generation. The measurements were taken in summer and winter and the relatively large variation is related to the seasonal variation in temperature and light. When only winter measurements were included, e.g., the length increment from one winter to the next was monitored, there was a perfect fit of the data to (1) (R2 = 1.00). Equation (1) also fits well to data from North Sea cod larger than 70 cm and is preferred over the 2nd order polynomial fit. However, North Sea cod with a length below 70 cm do not grow according to the prediction of (1). Since the farmed cod which was held at fairly constant conditions did grow according to (1), and that cod older than 0.5 years therefore seems to grow according to this relationship, it can be speculated that small North Sea cod encounter environmental conditions and food availability that are inferior to that of the larger cod.

In blue whiting and Pollock the best fit of the growth data was obtained with a second order polynomial model which was inverted compared to that in zebrafish (Fig. 1B). This could be interpreted as an indication that these species do not have an Lmax. However, it could also be an effect of varying environmental conditions in different life stages, where very young fish and spawning fish have more favorable conditions than juvenile fish. It was reported by Ianelli et al. (2011) that different size Walleye Pollock were found in different areas. Furthermore, data from both species had a relatively good fit to (1), with R2 = 0.89 and 0.95, respectively.

In (1), Lmax and k are individual characteristics determined by how the fish have adapted to the environmental conditions. Lmax is different from L∞ in von Bertalanffy’s growth function (von Bertalanffy, 1938); it is a fixed value, not an asymptote, it has a variance and it is not connected to time. Lmax is assumed to be genetically determined, and may change in response to long term changes in the environment and harvesting strategies, as described for North West Arctic cod (Borrell, 2013). It differs between species, between the stocks within a species and between the individuals within a stock. The different Lmax values for the two zebrafish strains discussed in this study (Gomez-Requeni et al., 2010; Kaushik, Georga & Koumoundouros, 2011) is an example of this variability. Clearly, there is a considerable number of variables that determine the value of k, and the parameterization will demand expertise within oceanography and in biological fields such as genetics and nutrition.

Equation (1) calculates length increment, while yield in biomass is the economically important variable. It has been shown that specific growth rate (SGR) in % of body weight per day has a linear relationship to fish weight on a log–log scale (Bjørnsson & Steinarsson, 2002; Braaten, 1984; Brett & Shelbourn, 1975; Iwama & Tautz, 1981; Jobling, 1983), which is an alternative formulation of the natural law of animal growth to (1). One reason for using length in our model is that both length of individual fish and number of fish can be measured directly using acoustic trawl surveys. The length increment data can then be converted to yield in weight and biomass using Fulton’s condition factor (Fulton, 1904). In aquaculture, (1) can be used to compare growth of individual fish and calculate yield of a standing stock of fish, both of which are related to k, and Lmax in different populations of fish can be determined.

According to (1), harvesting of fish which are at the point of maximum size increment is energetically most efficient. In the case of herring, fish of 10 cm length grow at a rate which is four times higher than fish of 30 cm length (Fig. 2B). Feed conversion efficiency generally increases with higher growth rates in fish (Kolstad, Grisdale-Hel & Gjerde, 2004), so utilization of the energy from plankton would be optimized by harvesting relatively small fish. The slaughter of farm animals prior to sexual maturation is a common practice in meat and fish farming which also takes advantage of the high growth rates in young animals. Selective fisheries on large fish may therefore defeat its own end, and prevent a maximum utilization of the marine resources. A similar perspective is presented by Borrell (2013), however the discussion of harvesting strategies has many aspects that need consideration beyond the scope of this study. Ecological principles may also be explained as optimizing the efficiency of energy utilization. An example is the seabirds along the Norwegian coast, which mainly feed on 0-group fish (Dragesund, Østvedt & Toresen, 2008), taking out fish that are in the phase of maximum energy conversion. This represents an efficient utilization of zooplankton biomass and energy.

Conclusion

The data presented here indicate that there is a natural law for length increment in fish, based on achieved length, which can be described by (1). Generally, data from young fish often deviate from the straight line obtained by fitting data from older fish to the model. This may be caused by varying environment conditions in different life stages of the wild fish. A consequence of (1) is that the efficiency of energy conversion from food decreases as the individual animal approaches Lmax. Length increment represents yield in biomass and energy and is the way living organisms store energy from the sun. Equation (1) enables us to quantify these processes.

We want to thank P Gomez-Requeni and I Rønnestad, the University of Bergen, Norway, for providing one of the datasets on Zebrafish and Ø. Karlsen, The Institute of Marine Research, Bergen, Norway, for the dataset on farmed cod. Thanks to Kjell Utne Rogn, The Institute of Marine Research for the blue whiting data, to Stan Kotwicki, the Alaska Fisheries Science Center, for data on Walleye Pollock, and Cecilie Kvamme, the Institute of Marine Research for tabulating the data on North Sea herring. Samuel J. Penglase is acknowledged for critical reading of the manuscript.

Additional Information and Declarations

Competing Interests

Author Contributions

Kristin Hamre is an Academic Editor for PeerJ.

Kristin Hamre conceived and designed the experiments, analyzed the data, wrote the paper.

Johannes Hamre conceived and designed the experiments, wrote the paper, formulated the growth model.

Espen Johnsen conceived and designed the experiments, wrote the paper.

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
