# Peer review of "A new model for simulating growth in fish"

_PeerJ, doi:10.7717/peerj.244_

## Round 0.1 · original submission · Major Revisions

The reviewers suggest major revision for this manuscript.

Reviewer 1 ·

Basic reporting

See "General Comments for the Author"

Experimental design

No Comments

Validity of the findings

Weak

Additional comments

This is an interesting paper that tries to use several sets of case study data to validate “a new fish growth model” (i.e. Equation 1). The preconditions for this model is that the length increment of fish living under constant conditions with sufficient food decreases linearly with fish length until it reaches zero at a Lmax. This is an easily understood and generally accepted concept. This model, if validated, will simplify the classical Beverton & Holt (B&H) model by excluding the variable of age. Although a few sets of the data fit well to the model, what I am concerned most is that the data used in this paper are not robust enough to support the conclusions reached. Firstly, zebrafish and mice data are from laboratory study, while data of cod, herring and capelin are either from field investigation or farmed fish. The wild or culturing environments and laboratory conditions may have substantial impacts on the growth (e.g. length and curves) of experimental animals. Using data derived from different environmental conditions to validate the new model and reach a general conclusion may be problematic. Secondly, growth of vertebrate is highly diverse in species. A good fit of the data from a specific mice study to the model cannot robustly support your conclusion that the law of length increment can be generalized to other vertebrates. Even in fish, data of different fish species, at least the groups of typical growth patterns, should be applied to validate the model before you conclude whether Equation 1 is valid or not. Lastly, it is hard to understand that Equation 1 can really describe the growth of fish as well as other vertebrate without considering the variable of age, because growth of animals is so different at life stages and is obviously a function of time.

Reviewer 2 ·

Basic reporting

The article meets the appropriate standard.

Experimental design

The topic is " A new model for simulating growth in fish", the observation in mice should be removed.

Validity of the findings

The results are reliable.

Additional comments

The manuscript is scientific validity and their suitability. It offers new theory to assess the growth in fishes.

---

## Round 0.2 · accepted · Accept

The reviewers and me think the manuscript has been improved and can be considered for publishing in the journal

Reviewer 1 ·

Basic reporting

I think that the manuscript has been improved and can be considered for publishing in the journal.

Experimental design

Please refer to the previous reviewing report

Validity of the findings

Please refer to the previous reviewing report